# Mapping Ecosystem Services in an Andean Water Supply Basin

**Diana Marcela Ruíz Ordoñez [1],*,† , Yineth Viviana Camacho De Angulo [1],† , Edgar Leonairo Pencué Fierro [2]**
**and Apolinar Figueroa Casas [1]**

1 Grupo de Estudios Ambientales (GEA), Environmental Sciencies, Department of Biology, University of Cauca, Calle 5 #4-70, Popayán 190003, Colombia
2 Optics and Laser Group, Environmental Sciences, Department of Physics, University of Cauca, Calle 5 #4-70, Popayán 190003, Colombia
* Correspondence: dianamruiz@unicauca.edu.co; Tel.: +57-317-3294094
† These authors contributed equally to this work.

**Abstract:** Socio-ecological dynamics affect the ecosystem services supply and are relevant to generate effective water management strategies; this condition is considered to evaluate under a holistic approach, the water ecosystem services (WES) in an Andean supply basin (ASB) in Colombia. This analysis focus on the connection of biophysical and sociocultural components for the multi-purpose use of water based on The Soil and Water Assessment Tool (SWAT) modelling for Las Piedras River Basin (LPRB). The generated Hydrological Response Units (HRUs), allows to estimate the capacity of the basin for supplying water (quantity) in adequate conditions (quality) for local populations in rural and urban areas, as well as WES zoning. The model was calibrated and validated to generate a baseline scenario, which was complemented with social cartography and participative workshops. The results indicate a low concentration of nitrogen and phosphorus, boosted by specific agro-ecological strategies developed by local communities; however, there are health risks for populations downstream and those that are supplied with water directly from the source. Additionally, Land Use and Land Cover (LULC) affects water availability, which demands restoration and conservation strategies to maintain WES supply for socioeconomic and cultural purposes, since different views on the available WES converge in the basin.

**Keywords:** ecosystem services supply; planning tool; water pollution; water supply; socioecological conflicts

## 1. Introduction

The use and ownership of natural resources to meet human needs and reach social wellbeing are concentrated in basins, which guarantee access to water as the enabling component for life, settlements, and economic-productive activities, such as agriculture. However, the interaction between LULC dynamics with climate variability influence the basin's capacity to supply continuous water in adequate quality for urban and rural communities. This transformation is relevant due to the socioecological conflicts that emerge when inequality and inequity for the availability of water (for drinking and household), widen the socio-economic gap, making water management strategies unpredictable and difficult to control.

In the Andean basins, this transformation comes mainly from productive activities that respond to raw materials' demand based on an economic growth model (capitalism), with known environmental liabilities. In the case of Colombia, livestock and agriculture are concentrated in Andean regions (Cauca and Magdalena basins), which in turn support more than 77% of the national population trough water supply and food production [1–4].

In the Upper Cauca River Basin (UCRB) in the Department of Cauca, the agriculture is characterized by small and medium scale crops, that extend toward the páramo (Andean moorland), affecting water sources, native vegetation, and soils, whose effects are intensified due to the uncertainty of climate change and local climate variability [5–8].

Agriculture practices in the UCRB includes deforestation, slash, burn, and the overuse of agrochemical inputs, that affects WES supply, limiting the water availability for local communities [9–11]. These conditions are represented in LPRB, a water supply basin in the southwest of Colombia, which provides multipurpose water for rural communities and drinking water for urban areas where socioecological conflicts around water management are presented.

These affectations to WES supply are related to productive activities, local climate variability and water demands, which have been studied from the biophysical and hydrological valuation approach independently, making analyses to estimate the effects on water supply, runoff dynamics, baseflow, flood events and peak discharges in basins, trough the LULC changes and water quality analysis, but not considering its socioecological integration for management and planning purposes [12,13], as is focused in this research.

Under these approaches, the main tools for analyzing WES are models that simulate the hydrological regulation dynamics based on LULC patterns, for this purpose the SWAT model has been worldwide used for assessing hydrological dynamics [14,15], to analyze freshwater supply and base flow conditions focused on the HRUs as well as the identification and zoning of WES based on hydrological response scenarios [16]. SWAT is also used for studying erosion processes [17], pollution by nutrients [18], basins management strategies [19,20], the monitoring of converting intensive agricultural practices to sustainable practices [21], or in the implementation of payment schemes for environmental services [22].

However, in Colombia, studies that include an integral analysis from a socioecological approach of WES with the SWAT model are not widely used, limiting the opportunities to generate planning tools for local governments.

One of these studies analyzed the climate change, LULC dynamics, and its effects on water yield and carbon sequestration in two Andean watersheds [23]; other research assessed the impacts of changing intensive tillage (IT) for conservation tillage (CT) in a potato crop. A study of the sediment yield, surface runoff and nutrient (nitrogen, phosphorus) losses in surface water runoff [24] was also used to evaluate the water yield in an Andean basin, where the SWAT model was used under different LULC and climate scenarios for water management [19], and in addition to evaluate the impact of LULC on the availability of water resources in conservation areas [25].

Although there are studies for SWAT model implementation at different scales in the country, just a few are developed in the ASB and less in the UPCR, where zoning of WES based on the hydrological dynamics, the relation with productive activities and climate variability are relevant, due to the socioecological dynamics that conditions the water supply for rural communities, increasing their vulnerability.

In this context, this study objective is to produce knowledge of ecosystem services based on hydrological dynamics in ASB, the socioecological conflicts related to WES supply, and the identification of zones for water management, using an integral methodology where participatory workshops and social cartography complements and validate the results of SWAT modeling.

The paper is organized as follows: First, we present an overview of the hydrological model and its application, the dataset and study area, the collection and processing of information. Second, we present the proposed tool for ecosystem services mapping in Andean basins, the results and analysis of applying the tool, and finally, we draw the corresponding conclusions and future developments.

## 2. Study Area

Las Piedras River basin (LPRB) is located within the municipalities of Popayán and Totoró in the UPCR in southwest of Colombia. It is at 76°31′10″ E and 2°21′45″ N. The basin is composed by two corregimientos (townships): Quintana and Las Piedras, where small peasants and indigenous communities (Puracé and Quintana councils) are located [26].

Figure 1 shows the study area, the river network, the weather stations, and land cover in the basin.

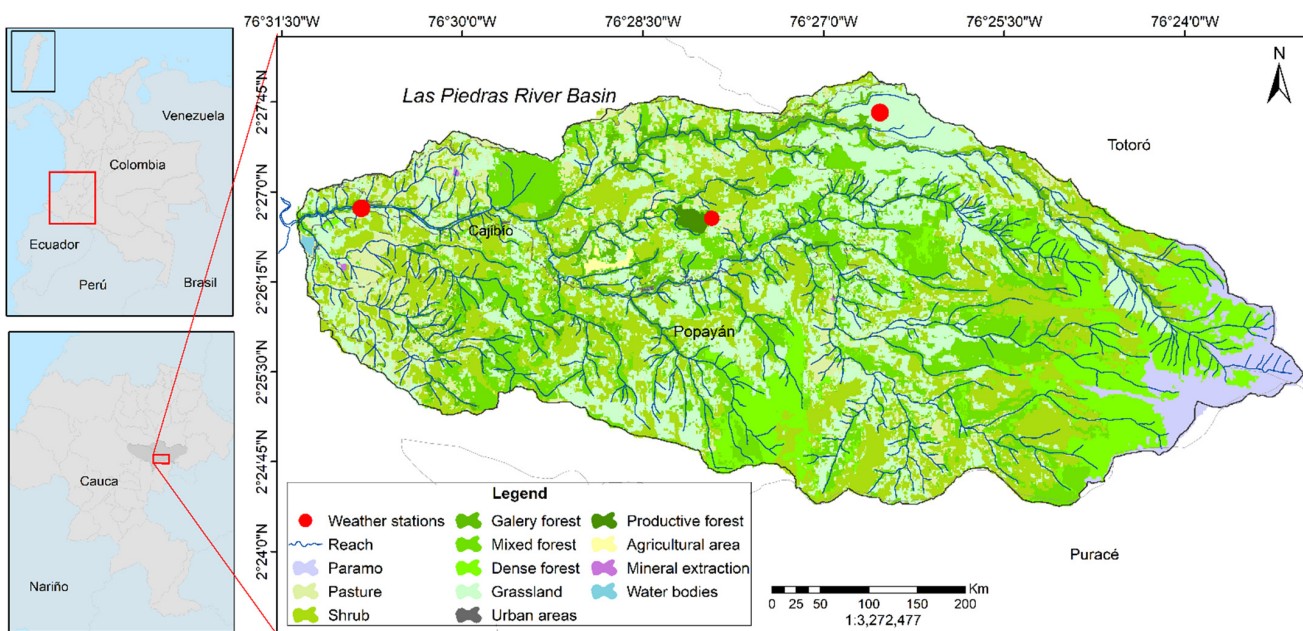

**Figure 1.** Study area, Las Piedras river basin (LPRB), Cauca, Colombia. In the figure the black framework corresponds to San Andres Island in the Colombian territory and the red framework extend indicator for department of Cauca.

## 3. Materials and Methods

The study was developed using the Method for Ecosystem Services Mapping (MESM) described in Figure 2. A proposed methodology based on mixed methods research which included (*i*) implementation of the SWAT hydrological model, based on updated cartographic inputs (*ii*) evaluation of ecosystem services (ES) supply complemented with social cartography to locate WES, and the (*iii*) analysis of ES distribution, to understand the socio-ecological conflicts that limit the availability of WES for the LPR communities.

**METHOD FOR ECOSYSTEM SERVICES MAPPING (MESM)**

**2. EVALUATION OF THE ECOSYSTEM SERVICES SUPPLY**

•**INPUTS**
 DEM, Land use map, Weather database, Type of soil map.
•**MODEL SET UP**
Delineating the basin and sub-basins, Creation and definition of the HRU, Climate generator and tables of meteorological data.
•**CALIBRATION AND VALIDATION**
Flow data (2000–2010), Participatory workshops, Field visits.

**1. IMPLEMENTATION OF THE SWAT MODEL**

•**IDENTIFICATION OF WES**
HRU priorization and zoning by subbasins
**PRIORITIZED ECOSYSTEM SERVICES**
Expert criteria workshop, Natural land cover (1–5), Deep slopes (1–3).
• **SOCIAL CARTOGRAPHY**
Jointly definition of ES and categories, List of ES, Locate ES on the map, Generate the WES map.

•**SOCIOECOLOGICAL CONFLICTS**
Anthropic land cover map vs hydrological ES, Participative workshops.

**3. ANALYSIS OF ECOSYSTEM SERVICES DISTRIBUTION**

**Figure 2.** Proposed method for ecosystem services mapping.

*3.1. Implementation of the SWAT Model*

SWAT is a dynamic and continuous model based on mathematical descriptions of physical, hydro-chemical, and bio-geo-chemical processes that combines elements of physical conditions and vegetation growth processes trough spatial disaggregation or HRUs. This model was developed by the Blackland Research Center in Texas in 1999 for the United States Department of Agriculture (USDA) [27]. SWAT models the basin and its dynamics based on different scenarios, using a semi-distributed deterministic model. It is useful for planning purposes due to the connection of different components of the territory, such as LULC, reforestation activities, population centers and catchment. The model is based on the water balance equation (shown in Equation (1)) to determine the input, output, and storage flows of water in the basin, as well as its hydric response.

$$SW_t = SW_0 + \sum R_{day} - Q_{surf} - E_a - W_{seep} - Q_{gw} \tag{1}$$

where $SW_t$ is the final soil water content (mm); $SW_0$ is the moisture content in one day $i$ (mm); $t$ is the time (days); $R_{day}$ is the daily precipitation of day $i$ (mm); $Q_{surf}$ is the surface run-off produced of day $i$ (mm); $E_a$ is the evaporation of day $i$ (mm); $W_{seep}$ is the content entering the vadose zone of the soil during day $i$ (mm); $Q_{gw}$ is the flow produced or returned of day $i$ (mm).

The database was created according to the objective of the study and the specific inputs requirements of SWAT model:

3.1.1. Inputs

- Digital elevation model (DEM): for topography, the study used a DEM with 12.5 m accuracy (cell size 12.5 × 12.5), obtained from the Alaska Satellite Facility website; the LPRB has an altitudinal gradient from 1980 to 3820 m.a.s.l.
- Land use map: the map contains information of the areas and landcover types present in LPRB. It was generated for April 2017 (low percentage of clouds) using images of the Sentinel 2A satellite platform, with 10 m precision, considering the Corine Land Cover methodology, adapted in Colombia and the algorithm developed by WP4 RICCLISA [28], identifying 14 landcover types from levels 1, 2, and 3. Field visits and key stakeholders' workshops validated this information (social cartography).
- Weather database: the database was generated from information available of daily precipitation data from nearby weather stations, for the period from 1 January 1999 to 31 December 2017. The statistical weather data required by the SWAT model are the multi-annual averages of maximum and minimum temperature and precipitation, standard deviation for each month, bias coefficient for daily precipitation, number of days of precipitation, probabilities of a humid day after a dry-humid day. These were calculated through the mathematical expressions suggested in the SWAT manual [27].
- Soil type map: contains information of the physical and chemical properties of the LPRB (scale 1:25.000), obtained from information on the study of soils by the planning and management document for LPRB [29].

3.1.2. Model Set Up

- Delineating the basin and sub-basins: The flow direction and the accumulation of water within the sub-basins was simulated with the inputs: DEM, mask of the study area, and the river network, as well as the definition of slope's range and the maximum and minimum elevations. Outlets were selected considering the main drains of the LPRB.
- Creation and definition of the hydrologic response units (HRUs): The HRUs map was based on the superposition of the shapefiles soil types (22 units), land use (12 types), and the specific slopes range (four ranges). From this output, a minimum percentage of aggregation was chosen by expert criteria, considering representative land use, soils, and slopes of the zone, allowing the prioritization of the HRUs, using 1% for LULC, 6% as minimum value for types of soils, and 10% for range of slope, with the

lowest loss of information over an area of the basin and the best distribution in the sub-basins [27].

- Weather generator and tables of meteorological data: Information was included based on the weather station identifiers and location of Arrayanales (ARR) and Diviso (DIV) stations daily precipitation database (mm) and its statistical data. Due to the lack of information in the study area, SWAT model was used to simulate and complete input values of solar radiation, relative humidity, and wind speed.

### 3.1.3. Calibration and Validation

The calibration of SWAT model for LPRB, was carried out through the SWAT—CUP (SWAT Calibration and Uncertainty Procedures) software with the SUFI2 algorithm [30], which operates through trial and error by randomly changing the values of parameters of interest, such as initial SCS CN II value (Cn2), base flow alpha factor (Alpha_Bf), Groundwater delay (Gw delay), threshold water depth in the shallow aquifer for flow (GWQMN), average slope steepness (slope), saturated hydraulic conductivity (Sol_K), among others. This is done until obtained a reasonable coincidence ($R^2 \geq 0.6$) between the simulation and the values observed.

The model was calibrated and validated with the daily precipitation data (1999–2017) from ARR and DIV weather stations and the monthly streamflow data (1999–2009) from the Puente Carretera (PCA) limnimetric station. Four iterations of 200 simulations each one was carried out, changing the parameters included in the SUFI2. The validation used registries of streamflow (2015–2016), with an iteration of 200 simulations. With the participative workshops and social cartography described in Section 3.2, we contrasted and validated the results from the SWAT model with social perception of the LPRB map.

### 3.2. *Evaluation of the Ecosystem Services Supply*

The evaluation of the ES supply was developed under a participative approach with experts and communities, through workshops, social cartography and field visits as established in the proposed MESM, to validate the hydrological modelling, prioritize the HRUs, proposed a joint definition of the concept of ecosystem services, their categories and for zoning each one of them under the land cover/slope combination.

### 3.2.1. Identification of WES

The WES identification was based on the hydrological conditions of the LPRB modelled with SWAT, with the resulting water-soil-climate-use-slope interaction for the 1999–2017 period, which is represented in the HRUs distribution map.

The SWAT outputs allow the analysis of (*i*) the water production as the water recharge (WYLD) and soil water availability (SW), potential and real evapotranspiration (ET-ETP), and surface runoff (SURQ). Additionally, (*ii*) for the estimation of water pollution, trough the variables of sediments yield and transport (SED YIELD), nitrates on surface runoff (NO3-SURQ) and organic phosphorus (ORGP)

From this, it is feasibly to group SWAT outputs by taking the sub-basin as the unit, to identify the water importance ones, due to the regulating function supported by natural land covers, and those in which is necessary to implement sustainable practices and soil management, considering the productive land covers.

### 3.2.2. Prioritized Ecosystem Services for LPRB

The HRUs were prioritized with experts and community's stakeholders participative workshops according to (*i*) dominant land cover, assigning importance values for hydrologic regulation from 1 to 5, the most important with one (1) score for natural coverages and the least important in regulation with five (5) score, for anthropic coverages; (*ii*) the slopes with greatest susceptibility to erosion processes [31] were assigned scores corresponding to the value of three (3) for the 0–25% range, a score of two (2) for the 25–75% range, and one (1) for the critical zones (75%).

### 3.2.3. Social Cartography

Social cartography is a participatory method that combines digital tools with qualitative methods to generate maps that represent the components, relationship and dynamics in specifics landscapes [32], in this case, we conducted workshops with communities of the upper, middle, and lower zones of the LPRB, to carried out four stages: (*i*) The first stage was to produced hand-drawn maps under the community perception to locate the main stream, tributaries, natural forest, crop production areas and pastures for livestock. These maps were then compared and complemented by the stakeholders with a press LULC map, this allowed us to validate the LULC map used as input for the SWAT model; (*ii*) the second stage was to create a jointly definition of ES and the corresponding categories of regulating, cultural and provisioning, then to list ES by each category; (*iii*) the third stage, was the location of each identified ES in the hand-drawn map using words, colors, pictures, or any other symbol to create the legend of the map; and in (*iv*) the fourth stage, the digitalization of this inputs into the HRUs map to generate the WES zoning.

### 3.3. Analysis of Ecosystem Services Distribution
Socioecological Conflicts

The water conflicts for local communities were analyzed trough the stakeholders (institutions and communities) perspectives of the water supply dynamics in LPRB, this was carried within a workshop to discuss about the different uses and views that each one of the stakeholders have respect to the LPRB as well as the action or strategy developed for its water management, this discussion was based on the results of SWAT model, the LUCL dynamics and WES zoning. The guiding questions for conducting the discussion about the use was: What do you use water for on your farm/home? For analyzing the views we ask the question: What does the LPRB represent for you, your family and/or community? For actions implemented the question was: What individual or community actions are developed to conserve water? Each question was discussed in focus groups of institutional, small peasants' and indigenous communities' representatives; the final output was a single statement that represents the collective thought.

## 4. Results

### 4.1. Implementation of the SWAT Model

This section presents the updated cartographic outputs for LPRB, as a result of the SWAT modelling, such as the total area of the basin, the tributaries, the calibration parameters, and the hydrograph.

### 4.2. Updated Outputs of the SWAT Modelling of LPRB

The LPRB delimitation was updated with a total area of 6606.27 ha, approximately 20 ha less than the one reported by [26], due to the precision of the DEM used in the study, from 30 m to 12.5 m. Based on this, 18 sub-basins were identified in the LPRB, compared to 13 reported by [33], regarding to this, the Robles sub-basin is included (6), the Santa Teresa sub-basin is divided into Santa Teresa (2), Las Pavas (3) and Santa Teresa II (4), the Aguas Claras sub-basin is divided into Aguas Claras (13), La Cabaña (14) and San Pedro (15) And the Buena Vista sub-basin is divided into the El Cedro (17), Peñas Blancas (18) and Piedra Negra (16), as shown in Figure 3 for details and comparisons.

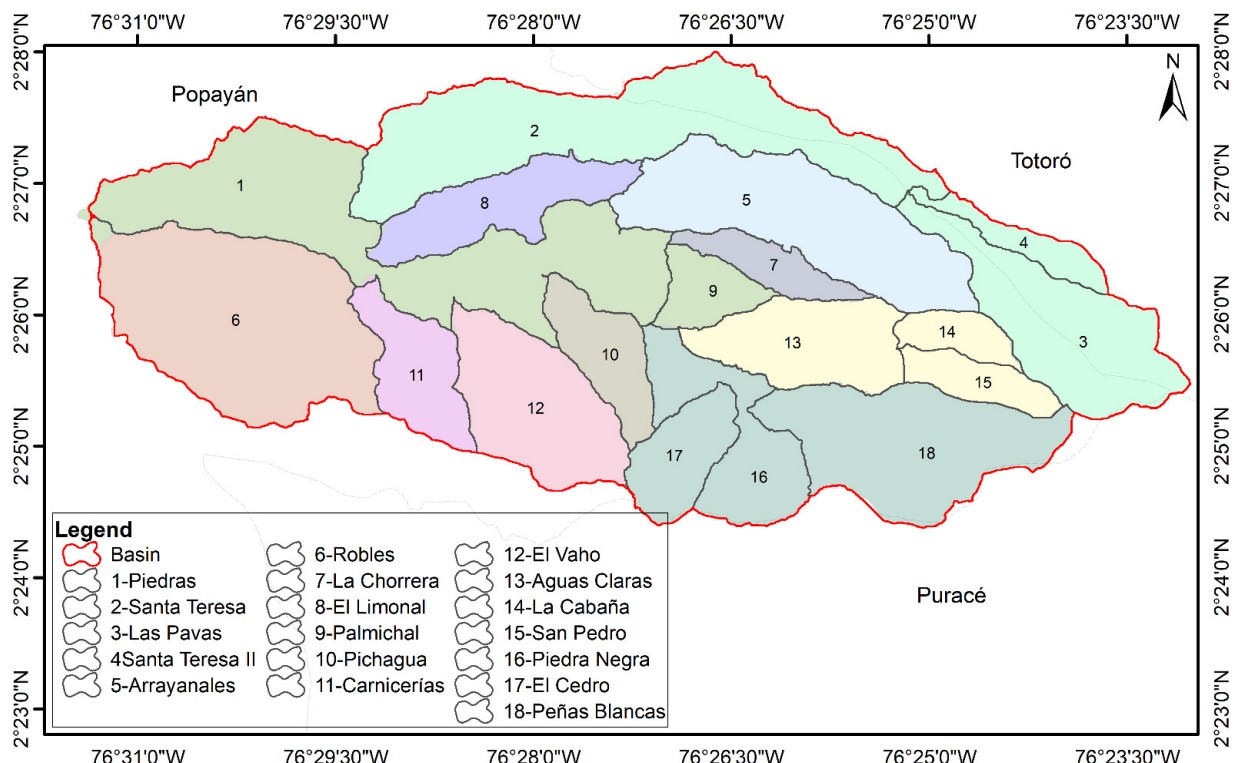

**Figure 3.** Sub-basins obtained with SWAT (delineated and numbered), compared with the sub-basins reported by [33] (differentiated by colors).

### 4.3. SWAT Calibration and Validation

The values of the determination coefficient were $R^2 = 0.99$ for ARR and $R^2 = 1$ for DIV. Water quality data were not considered in the calibration because these do not complement the minimum historical data, however, satisfactory calibration was obtained with $R^2 = 0.614$ [34] for monthly streamflow data (Figure 4), Table 1 shows the calibrated values for the parameters of interest.

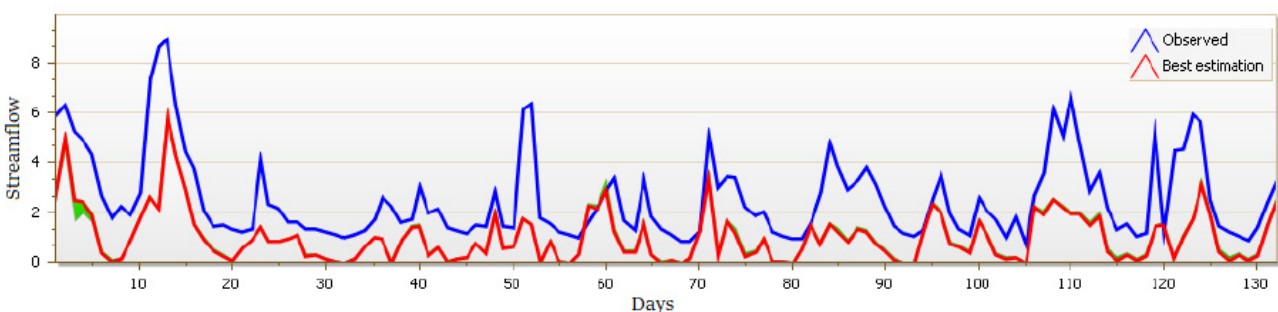

**Figure 4.** Simulated and observed streamflow hydrograph.

**Table 1.** Calibration parameters of the SWAT model for LPRB.

| ID | Parameter | Description | Process | Initial Range | Calibrated Value |
|----|-----------|-------------|---------|---------------|------------------|
| 1 | GWQMN (Threshold water depth in the shallow aquifer for flow) | Threshold of water depth | Base flow | 550–1000 | 862.96 |
| 2 | Alpha-Bf (Base flow alpha factor) | Base flow factor | Base flow | 0–1 | 0.5 |
| 3 | Gw-Delay (Groundwater delay) | Storage of groundwater | Base flow | 0–50 | 26.86 |
| 4 | Cn2 (Initial SCS CN II value) | | Run-off | 35–98 | 45.63 |

### 4.4. Evaluation of the ES Supply

This section presents the identification of WES as result of the hydrological modelling, which stablishes the baseline conditions for water supply in LPRB, the identification and the prioritization of WES according to the stakeholders' perspectives and the zoning of WES under the communities' views.

### 4.5. Identification of WES

A 607 HRUs map of the LPRB was prioritized (from an initial 1687 HRU map) by a minimum percentage of aggregation considering representative land use, soils, and slopes of the zone, covering 100% of the basin modeled with the best distribution in the 18 sub-basins [27], from this output. The actual condition of the LPRB was modelled made at sub-basins level; the results are shown in Figure 5.

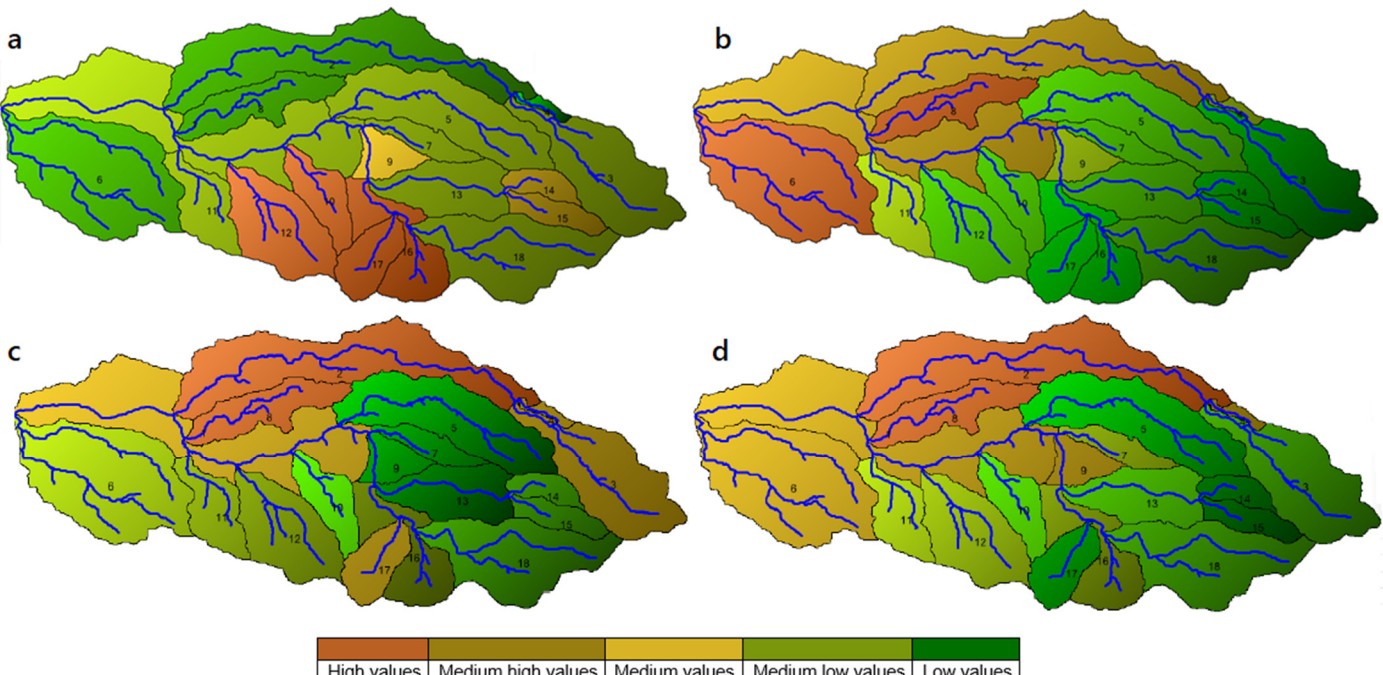

**Figure 5.** Hydrological simulation of LPRB. (**a**) Water recharge (WYLD). (**b**) Amount of available soil water (SW). (**c**) Sediment yield (SED YIELD). (**d**) Nitrates on surface run-off (NO3-SURQ).

The water production in the LPRB, i.e., is represented by the amount of water generated in each sub-basin reaching the streamflow, Figure 5a,b, shows the offer of WES by the parameters WYLD and SW, corresponding to water balance in the land phase [35,36]. Thus, it was possible to identify the sub-basins with the highest water contribution (WYLD), corresponding to areas with prevalence of high slopes (≥75%) and natural land cover (dense forest, páramo, and shrub (Figure 1) such as the sub-basins 17 (856.92 mm), 3 (664.13 mm), 10 (818.56 mm), 16 (808.83 mm), 12 (798.89 mm), and 14 (740.54 mm).

The green color identifies the sub-basins with the lowest water recharge, which are areas with crops and grasslands (clean and degraded), in the 4, 2, 8, 6, and 1 sub-basins. The amount of water stored in the soil (SW) for plants, increases toward lower zones, especially in sub-basin 2 (748.13 mm), 8 (803.49 mm) and 6 (872.06 mm), corresponding to productive areas. The water losses from soil surface in the LPRB, were analyzed under weather conditions simulated with the Penman–Monteith method (1999–2017), for the ET and ETP. Natural covers of the upper area had high values of ETP in sub-basin 3 (3500 mm), 15 (3497 mm) and 18 (3473 mm) while crops and pastures of the lower area presented high values of ET in sub-basins 2 (947.69 mm), 8 (942.95 mm), 6 (936.52 mm) and 9 (827.63 mm). The water consumption by forests is greater than in other vegetation

types due to the depth of roots, height, and foliage. Water is retained and stored in the soil-vegetation interphase and regulates source recharge processes, while zones with higher ET are susceptible to drought due to poor retention and regulation of soil water [37]. In these dynamics, weather conditions, productive and management practices are determinant for hydrological regulation, in the case of the LPRB, the areas with dominant agricultural and fish farming activities are sub-basins with high ET values.

With respect to water quality, in the LPRB, the SED YIELD parameter shown in Figure 5c resembles the nutrient loss response, where the sub-basins 2 (0.88 ton/ha), 8 (0.913 ton/ha) and 1 (0.49 ton/ha) presented higher accumulation of contaminants in the soil (in brown) from agro-chemicals of the potato's crops.

According to the NO3-SURQ parameter, the largest yields were presented in sub-basins 8 (1.11 kg/ha), 2 (1.06 kg/ha), 4 (0.98 kg/ha) and 1 (0.81 kg/ha), related to the use of agro-chemicals for crops and livestock production Figure 5d. The sub-basins 1 (0.81 kg/ha), 9 (0.79 kg/ha), 13 (0.44 kg/ha), and 14 (0.32 kg/ha), had lower crop production because these are conservation areas with fragile ecosystems and low-fertility soils. The results showed a similar behavior for phosphorous and nitrogen, the largest producers of ORGP were the 1 (1.57 kg/ha), 2 (3.41 kg/ha), 8 (3.94 kg/ha) and 11 (1.40 kg/ha) sub-basins, related to the larger monocrop and livestock production zones. The distribution of this nutrient is key in management processes because it comes from both organic (ash, manure) and chemical (commercial) sources, and is enhanced in scenarios of excessive fertilization combined with soil compaction caused by cattle trampling.

### 4.6. Prioritized Ecosystem Services for LPRB

The joint definition of ES established by local communities of the LPRB is as follows: "*Ecosystem services are what nature provides to people, it results from interaction with human beings, where man receives benefits*". To understand the specific categories of ES, communities in LPR established that regulating ES represents: "*Equilibrium in the biological processes of ecosystems*"; the cultural ES are: "*Goods and materials that contribute to inner wealth*"; and finally, the provisioning ES refer to: "*What nature gives us*". The specific WES identified and classified by the upper, middle, and lower zone of the LPR, are presented in Table 2.

**Table 2.** Prioritized ecosystem services for LPRB.

| ES CATEGORY | PRIORITIZED ES | | |
| --- | --- | --- | --- |
| | **Upper** | **Middle** | **Lower** |
| Provisioning | ▪ Food sovereignty <br> ▪ Water availability for the communities | ▪ Water quantity <br> ▪ Water quality | ▪ Productivity availability of timber resources <br> ▪ Good quality water |
| Regulating | ▪ Air regulation <br> ▪ Climate regulation <br> ▪ Hydrological regulation | ▪ Oxygen availability <br> ▪ Biological control <br> ▪ Climate regulation | ▪ Soil nutrient cycling <br> ▪ Pollination <br> ▪ Oxygen availability |
| Cultural | ▪ Sacred sites <br> ▪ Maintenance of oral tradition <br> ▪ Knowledge of the territory | ▪ Traditional knowledge <br> ▪ Maintenance of oral tradition. | ▪ Field schools <br> ▪ Ecotourism areas <br> ▪ Traditional knowledge of the territory <br> ▪ Network of civil society reserves for conservation |

### 4.7. Social Cartography

According to the WES map (Figure 6) the upper zone of the LPRB was related mainly with regulation and cultural ES, associated with zones of natural regulating coverages that are identified as sacred or pilgrimage sites like the Puzná Mountain. The middle zone had an important supply of cultural ES related with property appraisals and the ecotourism potential, because of their strategic location toward zones with high slopes, conservation,

and areas of environmental protection. The lower zone represented the availability of provisioning ES, associated in this case with grazing and crop areas.

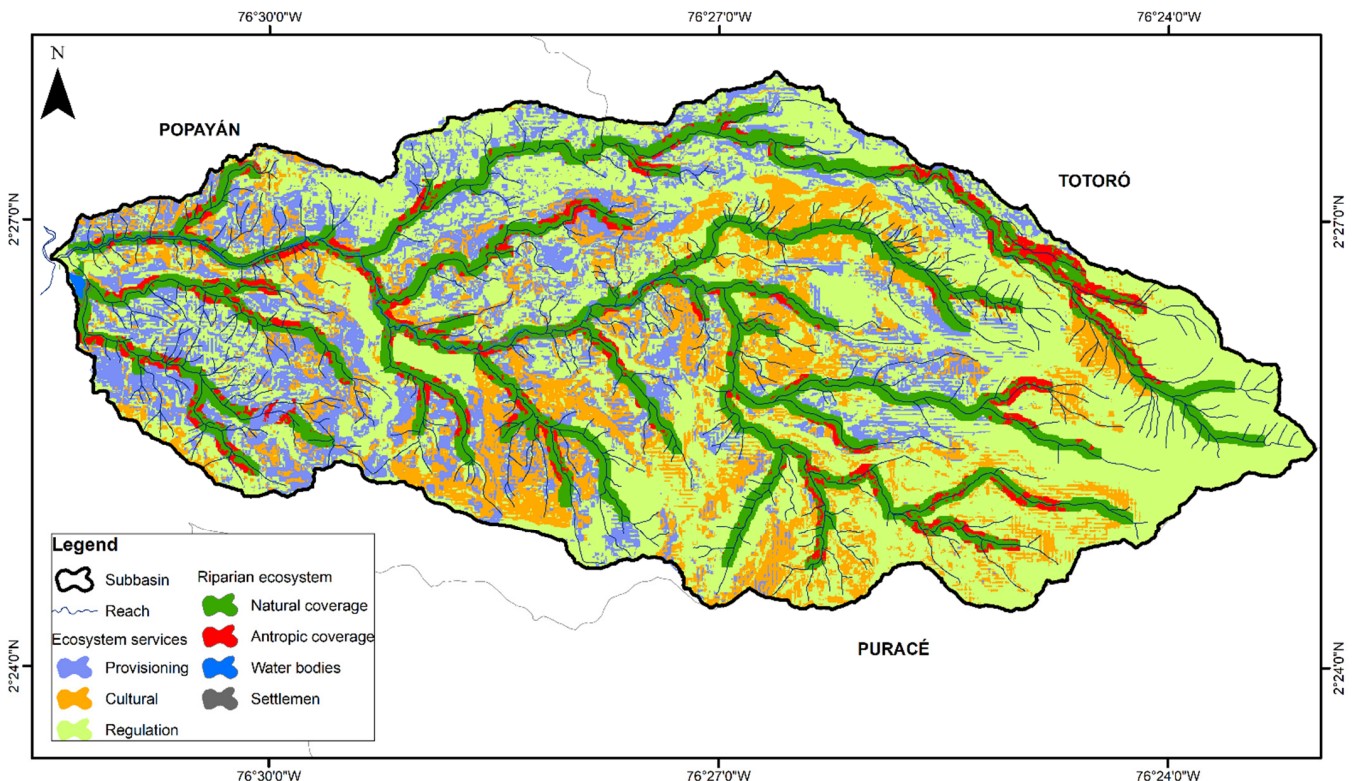

**Figure 6.** Distribution of WES in the LPRB.

The ES analysis evidenced the community relationships with their territory trough a deep connection with the LPRB "*the water connects us*", where the provider and beneficiary stakeholders' dynamics, as well as the productive activities developed, condition the opportunities to sustainable socioecological transition processes.

The ES approach has been incorporated into the management strategies of some local entities, such as the ES payments in farmers with water recharge areas, which includes a property tax discount by the municipal administration (Agreement 30/2012) and environmental educational processes. These sorts of payments are applicable to rural properties located in areas with hydrological importance for the water intake of the municipal water service, that has been recognized by the environmental authority. This management strategy is included in the planning and environmental conservation processes of the municipal aqueduct enterprise. From local communities' leadership, an important strategy for conservation is the creation of civil society's nature reserves (a formal protection figure recognized by Colombian environmental ministry) and the natural reserves that still are not recognized by local government, which in turn, constitutes important places for ecotourism routes.

### 4.8. Sociecological Conflicts

The production areas established in the riparian buffer zone (Figure 6), represent an important source of trade-offs between ES, such as food supply, the regulation of water quality, barriers of sediments, and nutrients from hillside areas, where livestock has established. Additionally, these crops have strong exposure to the effects of prolonged periods of rainfall, LPRB communities indicated that drought effect is exacerbate by the soils' low fertility and steep slopes, causing socioeconomic affectation to the families, that depends entirely on the agriculture and livestock.

These productive activities on the riparian and hillside area were relevant in the sub-basins of the upper zone of the LPRB, such as sub-basins 2, 5, and 8, with extended

potato, bean, and corn crop areas, which are produced under conventional schemes with agrochemical inputs, logging, and burning practices. Although the main production activity in the LPRB is livestock, conventional agriculture practices and fish farming are important sources of pollution, as the SWAT model reveals with respect to the areas with this type of activities. Thus, these production processes limit the supply of WES, due to affectations on water quality and quantity.

From the stakeholder's analysis, it was possible to identify the key points of divergences and convergences with respect to their views of the LPRB, as well as the management actions taken by each one of them under these views. That is, the institutional actors consider the LPRB as a water production space for drinking water; for small peasants, it represents the territory for productive activities and multipurpose water supply that supports their socioeconomic needs, and for indigenous communities, it is the space for silvopastoral systems and multipurpose water supply, where community coexist with nature in an ancient right where the territorial expansion is needed. Due to these different views of the LPRB, and despite the peace and convivence agreement signed between local communities, it has been difficult to articulate the water management strategies; some of them are presented in Figure 7.

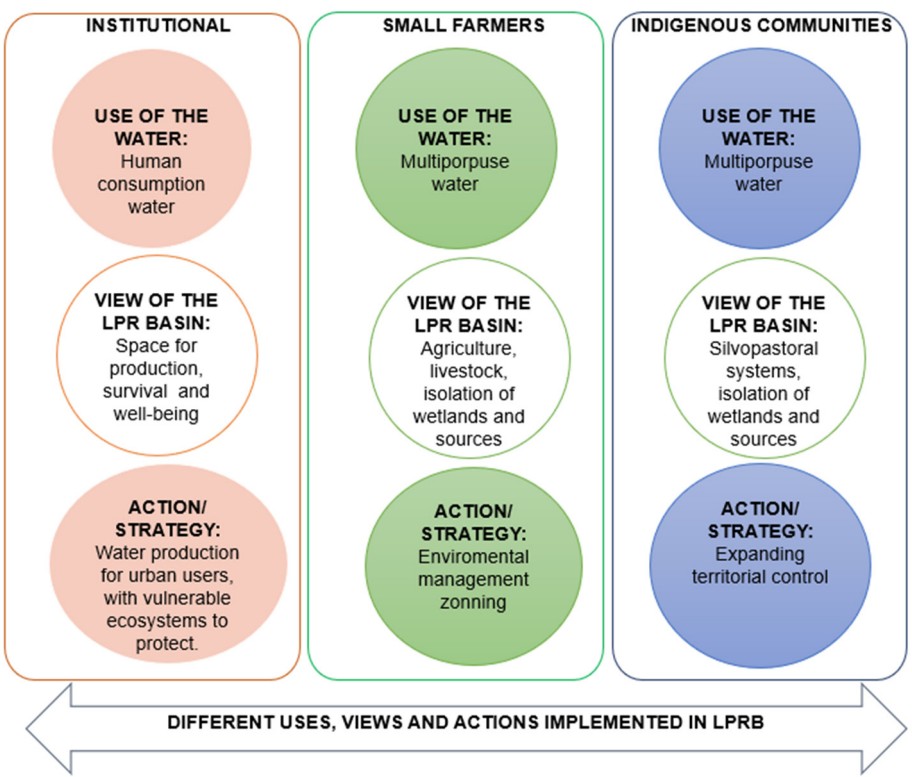

**Figure 7.** Different stakeholders' perspectives of the water supply in LPR.

## 5. Discussion

One of the most important contributions of this study is the possibility of performing WES supply analysis in watersheds of water importance, such as those of the Andean zone, which do not have detailed or historical inputs for the implementation of hydrological models. The MESM offers a methodological integration that draws on the strengths of quantitative and qualitative methods.

### 5.1. Implementation of the SWAT Model

The resolution of the DEM (12.5 m) improves the landscape shapes, making the delimitation of the LPRB contour more precise with an area approximately 20 ha less than reported in previous studies [26,33,38], as well as the identification of the 18 sub-basins, six

(6) more than reported by [38] and seven (7) more than reported by [33]; additionally, we confirm that sub-basin 6 is included in the LPRB area, as reported by [33].

For calibration, the monthly values streamflow observed and the best estimated shown in Figure 4; corresponds to January 1999 and December 2010 ($R^2$ = 0.614), The most distant values are because the simulation was made without taking into account the macro-scale weather conditions, as the Southern Oscillation (ENSO) with the warming phase or Niño and the cold phase or Niña, due to the missing historical data; in the region the ENSO was presented during five years (1999, 2000, 2007, 2008, 2011). This caused an underestimation in the simulated streamflow, so it is relevant to mention that the availability of climate data with a minimum historical record of 15 years is important to improve the calibration of the model and to compensate for the lack of climate information [34]. The SWAT model validation was complemented with social cartography and participatory workshops; this allows to improve the LULC map, one of the most important inputs for SWAT, as well as providing a space for knowledge dialogue with stakeholders.

Through the HRUs map, is possible to identify that the largest extension of HRUs corresponds to anthropized cover (grassland) related to livestock, the main productive activity in the LPRB [33]. The natural regulating land covers, such as dense forest, is low (11.55% of the sub-basin area), located in the higher zones, where agricultural and livestock is limited by the conditions of the terrain with pronounced slopes and hillside areas, this corresponds to conservation areas isolated by stakeholders The natural pasture cover of the entire LPRB, is an area of socioeconomic interest because it is a potential area for expanding urban and productive activities.

*5.2. Evaluation of the ES Supply*

The WES related to water quantity, represented by the parameters WYLD and SW, indicate that the sub-basins with the higher water supply (WYLD) are located towards the upper areas, where there is less presence of productive activities, the sub-basins with the greatest amount of water stored in the soil (SW) are located in areas with soils with moderate agrological capacity, where medium-scale productive practices are developed. The ET is higher in the sub-basins of the middle and lower zones, due to the cultivated areas, while in the upper zone, the ETP is higher due to the evaporation processes of rainfall intercepted by the canopy and tree transpiration of the paramo [37].

The sub-basins that contribute the greatest concentrations of nutrients and sediments are Limonal (8), Santa Teresa (2), Santa Teresa II (4), and Las Piedras (1); these areas have productive activities, but soils have moderate to low fertility and agrological capacity, which demands the implementation of sustainable production systems, especially in upper sub-basins 2 and 4.

To estimate the dynamics of the WES related to quality, we analyze the N and P cycle associated with production practices in the sub-basin which are low, since the use of agrochemicals is not generalized and there are transitions towards the use of organic agro-inputs; however, this condition is not common to the supply sources of the department of Cauca or the Andean zone [10,17]. Although the values calculated are low with respect to the whole basin area, these processes are directly related with the water quality changes of the source basin for the municipal water service, affecting the potabilization processes required for urban users, increasing fees, continuity, and quality of the water supply [19,23].

In this sense, the supply of WES is related to the needs of the communities in the availability of water and food for the LPRB inhabitants and for human consumption in urban areas, but in both cases, there are health risks for populations, on the one hand by drinking water directly from the source and on the other hand by the potabilization requirements. From the socioecological view of the LPRB, we identify problems with access to water for LPRB inhabitants, by quantity (high and medium zones) and quality (low zone) and although the agricultural production is in the process of converting to sustainable practices, the fertility limitations that characterize the LPRB soils must be overcome, and the commercialization channels must be improved.

*5.3. Analysis of ES Distribution*

Community groups of the LPRB evidence their differences through the prioritization of cultural ES; in the upper zone (mainly indigenous communities) we identify ES related to knowledge of the territory and the collective cultural heritage. In the lower zone, organizational and associative activities of the small farmers communities prevailed, and in the middle zone, we find a transition between the two indigenous-small peasants' visions, by prioritizing aspects that give greater "value" to their lands in the sense of environmental importance with ecotourism areas [13].

Because of this, it is necessary to strengthen the social network through the articulation of community and institutional stakeholders. To generate synergies for conducting planning and management processes, such actions must be accompanied by strategies for improving socioeconomic conditions of the local inhabitants, where their broad knowledge and environmental sense could be included in community-based productive alternatives that allow WES supply and better-quality life.

## 6. Conclusions

In this paper, we present an integrated analysis of the supply of water ecosystem services in a strategic Andean water supply basin in Colombia. The results show the high susceptibility to hydric erosion due to changes in texture and structure of the soils in the LPRB, which is the result of continuous implementation of agricultural activities with inadequate technologies. This condition affects the availability of nutrients, generates loss of soil fertility, and increases run-off rates, related, in turn, with dynamics in nutrient concentration and alteration of the pH in water from the stream. Additionally, the middle and lower zones of the LPRB are in drought risk due to the poor retention and regulation of water in the soil, as indicated by high ET values associated with crop areas.

According to the SWAT model, we identify the sub-basins that demand restoration and conservation actions, because of their hydrological importance: Las Pavas (3), Pichagua (10), and San Pedro (15), as well as the sub-basins where it is necessary to strengthen the processes of soil management as they represent areas with a predominant anthropic land cover, Santa Teresa (2), Limonal (8), and Cedro (17), and productive activities for sustain local communities. Additionally, the sediment production and transport in the LPRB is higher in the Santa Teresa (2), Limonal (8), and Piedras (1) sub-basins, related to agrochemicals, used for potatoes' crops, while the Arrayanales (5) and La Chorrera (7) sub-basins show low sediment accumulation, because these are conservation areas delimited by stakeholders.

As future developments, we would like to consider the modelling of management scenarios with: (i) diffuse pollution processes and soil compaction, in relation to the main productive activities developed in the basin that are affecting hydrological ecosystems services; and (ii) the interactions between inhabitants of the LPRB in the rural area and uses of the water supply system in the urban area.

**Author Contributions:** Conceptualization, methodology, validation, formal analysis, writing—review and editing D.M.R.O. and Y.V.C.D.A.; resources, supervision, review and editing E.L.P.F.; writing—review and editing A.F.C. All authors have read and agreed to the published version of the manuscript.

**Funding:** The authors would like to thank the different entities involved in providing the funding to develop this research work. This work was supported by Universidad del Cauca (501100005682) under Grant number (BPIN 2020000100714) ID 5650 and ID 5142 supported by the Water Security and Sustainable Development Hub funded by the UK Research and Innovation's Global Challenges Research Fund (GCRF) [grant number: ES/S008179/1]. this work was performed with the support of the Department of Science Technology and Innovation Colciencias in Colombia, through the bicentennial scholarship program "Young research and doctoral students—567/2012".

**Institutional Review Board Statement:** The study was conducted in accordance with the Declaration of Helsinki, and approved by the Ethics Committee of the University of Cauca (Protocol 6.1–1.25/23, 2 December 2020).

**Informed Consent Statement:** Informed consent was obtained from all subjects involved in the study.

**Data Availability Statement:** Not applicable.

**Acknowledgments:** This research was performed with support from the Ministry of Science Technology and Innovation, MINCIENCIAS, in Colombia, through the Bicentennial Scholarship Program "Young researchers and doctoral students", the Water Security and Sustainable Development Hub funded by the UK Research and Innovation's Global Challenges Research Fund (GCRF) (grant number: ES/S008179/1) and the project "Vulnerability and Risk in Drinking Water Systems in Cauca—AQUARISC" financed by the science ministry in Colombia. Gratitude is expressed to the Las Piedras River Foundation, the Environmental Studies and Optical and Laser Research Groups at Universidad del Cauca. The authors also thank the indigenous and small farming communities living in the LPR basins and Doctor Tatiana Solano for her review and contributions.

**Conflicts of Interest:** The authors declare no conflict of interest.

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
