# Peer review of "Mapping Ecosystem Services in an Andean Water Supply Basin"

_sustainability, doi:10.3390/su15031793_

Round 1
Reviewer 1 Report
In this work, an integrated analysis of the supply of water ecosystem services in a strategic Andean water supply basin in Colombia was conducted. The methods used are persuasive and results are convincing. However, the formatting is not good.
1. Figure 2 and 6 are not clear.
2. The formatting should be carefully revised. Many references are not clearly marked in the manu.
3. The title of Figure 3 is not written.
4. Languages in Figure 6 should be revised.
Author Response
The authors are grateful to the Associate Editor and all the Reviewers for the constructive criticisms that allowed a significant improvement of the quality of this manuscript. In the following, detailed replies to all the comments are provided. Revisions to the manuscript was marked up using the “Track Changes” as requested. The detailed response is presented in the attached file.

Reviewer 2 Report
Article number= sustainability-2095448.
1. The title seems ok.
- The abstract seems to be good. Please add one more introductory line of your objective in beginning of abstract.
- Research gap should be delivered on more clear way with directed necessity for the future research work.
- Introduction section must be written on more quality way, i.e., more up-to-date references addressed.
- The novelty of the work must be clearly addressed and discussed, compare previous research with existing research findings and highlight novelty.
- Please check the abbreviations of words throughout the article. All should be consistent.
- What is problem statement?
- The main objective of the work must be written on the more clear and more concise way at the end of introduction section.
- Please provide space between number and units. Please revise your paper accordingly since some issue occurs on several spots in the paper.
- Overall result section is well explained but the arrangement is very weird.
- Please add a comparative discussion section. It would be more better for reader.
- Conclusion and Future perspectives should be added in section 5. Conclusion section is missing some perspective related to the future research work, quantify main research findings, highlight relevance of the work with respect to the field aspect.
- To avoid grammar and linguistic mistakes, major level English language should be thoroughly checked. Please revise your paper accordingly since several language issue occurs on several spots in the paper.
- Reference formatting need carefully revision. All must be consistent in one formate. Please follow the journal guidelines.
- Please follow the MDPI guide lines to prepare your articles. It was very difficult to read and understand. Also added a concluding remarks.
Decision = Moderate level revision required.
Author Response

(The authors gave the same response as above.)

Reviewer 3 Report
This is a very interesting manuscript trying to map ECOSYSTEM SERVICES using SWAT model. There are few suggestions which hopefully can improve the manuscript.
In the introduction section, there are too many paragraphs.
There are language problems: in line 57 “conducting analyses”, line 191 “stablished”, line 268 “1687 HRUs”, please check the whole manuscript carefully.
In figure 1, it is not the DEM as stated in line 100. The location of hydrological stations (Puente Carretera limnimetric station) and meteorological stations should be indicated in this figure. The name of the subbasin could be showed on the map instead of the legend.
In figure 1 it should be weather generator instead of climate generator.
Several references were not properly organized in line 99, 105, 243, 257, 276, 278, 282 and many other places.
Detailed calibration and validation information should be included such as the simulated and observed hydrograph. The value 0.6 in line 180 is not clear, is it the NSE or R2? I am afraid that more detailed information about the participative workshops and social mapping need to be provided.
For identification of WES in section 3.2, the process is not clear at the moment, you can use an example or figures to better explain the idea. The detail of social cartography workshops is not clear, what have been done? How it is done is note clear at the moment.
The calibration and validation should be mentioned only once instead only once instead of twice in line 173 and 248.
Line 268 to 273 should be mentioned in the model development section not here.
There are some blank (white) pixels in figure 5, what does these mean, it is not explained in the legend.
Author Response

(The authors gave the same response as above.)

Round 2
Reviewer 1 Report
Good